# Molecular Epidemiology of *Xanthomonas*
*euvesicatoria* Strains from the Balkan Peninsula Revealed by a New Multiple-Locus Variable-Number Tandem-Repeat Analysis Scheme

**DOI:** 10.3390/microorganisms9030536

**Published:** 2021-03-05

**Authors:** Taca Vancheva, Nevena Bogatzevska, Penka Moncheva, Sasa Mitrev, Christian Vernière, Ralf Koebnik

**Affiliations:** 1IPME, Univ Montpellier, Cirad, IRD, Montpellier, France; tacavancheva@gmail.com; 2Department of General and Industrial Microbiology, Faculty of Biology, Sofia University ‘St. Kliment Ohridski’, Sofia, Bulgaria; montcheva@biofac.uni-sofia.bg; 3Institute of Soil Science, Agrotechnologies and Plant Protection ‘Nikola Pushkarov’, Sofia, Bulgaria; nbogatzevska@abv.bg; 4Department for Plant and Environment Protection, Faculty of Agriculture, Goce Delchev University, Štip, North Macedonia; sasa.mitrev@ugd.edu.mk; 5Plant Health Institute of Montpellier (PHIM), Univ Montpellier, Cirad, INRAe, Insitut Agro, IRD, Montpellier, France; christian.verniere@cirad.fr

**Keywords:** bacterial spot, genetic diversity, molecular typing, pepper, tomato

## Abstract

Bacterial spot of pepper and tomato is caused by at least three species of *Xanthomonas*, among them two pathovars of *Xanthomonas euvesicatoria*, which are responsible for significant yield losses on all continents. In order to trace back the spread of bacterial spot pathogens within and among countries, we developed the first multilocus variable number of tandem repeat analyses (MLVA) scheme for pepper- and tomato-pathogenic strains of *X. euvesicatoria*. In this work, we assessed the repeat numbers by DNA sequencing of 16 tandem repeat loci and applied this new tool to analyse a representative set of 88 *X. euvesicatoria* pepper strains from Bulgaria and North Macedonia. The MLVA-16 scheme resulted in a Hunter–Gaston Discriminatory Index (HGDI) score of 0.944 and allowed to resolve 36 MLVA haplotypes (MTs), thus demonstrating its suitability for high-resolution molecular typing. Strains from the different regions of Bulgaria and North Macedonia were found to be widespread in genetically distant clonal complexes or singletons. Sequence types of the variable number of tandem repeats (VNTR) amplicons revealed cases of size homoplasy and suggested the coexistence of different populations and different introduction events. The large geographical distribution of MTs and the existence of epidemiologically closely related strains in different regions and countries suggest long dispersal of strains on pepper in this area.

## 1. Introduction

Bacterial spot, which is caused by at least three *Xanthomonas* species, is a threatening disease of pepper and tomato plants worldwide [1,2,3]. Under favourable climatic conditions, the pathogens can cause significant yield losses in pepper- and tomato-growing areas. Bacteria enter via natural openings and colonize the apoplast. The disease is characterized by small, irregular, water-soaked, greasy-appearing lesions on all above-ground parts of the plants, followed by cell death, tissue necrosis and egress of *Xanthomonas* to the leaf surface [4]. Defoliation and shedding of fruits and blossoms are commonly observed in pepper production areas. This leads to a reduction in fruit quality and fruit loss due to the presence of lesions and the proliferation of secondary postharvest pathogens [5].

A wide range of genetic and physiological variation was found among *Xanthomonas* strains that cause bacterial spot on tomatoes and peppers. Until the early 1990s, the causal agent of bacterial spot on pepper and tomato was *Xanthomonas campestris* pv. *vesicatoria*. In 1994, Stall et al. identified two genetically distinct groups of strains, referred to as A and B strains, within a worldwide collection based on genotypic and phenotypic characteristics [6]. Vauterin et al. proposed reclassification of the bacterial spot agents and divided *X. campestris* pv. *vesicatoria* into two species, with phenotypic group-A strains placed to *Xanthomonas axonopodis* pv. *vesicatoria* and phenotypic group-B strains relegated to *Xanthomonas vesicatoria*, as initially proposed by Dowson in 1939 [7,8]. In 2000, restriction fragment length polymorphisms and DNA sequence information of the 16S rRNA and adjacent intergenic sequence allowed to define two new pepper- and/or tomato-pathogenic *Xanthomonas* groups, C and D, with the D-group strains assigned to the species level *Xanthomonas gardneri* [9,10], which had been introduced in 1966 by Dye [11]. Based on DNA-DNA hybridization experiments, Jones et al. suggested in 2004 to consider the C-group strains as a novel species, called *Xanthomonas perforans* [12]. Only recently, genome-wide nucleotide sequence comparisons revealed that *X. perforans* does not deserve species status and was reclassified as a pathovar of *Xanthomonas euvesicatoria*, thus belonging to the same species as the previous group-A strains, also known as *X. axonopodis* pv. *vesicatoria*, which were reclassified as *X. euvesicatoria* pv. *euvesicatoria* [13,14,15]. Finally, based on whole-genome analyses, group-D strains, also known as *X. gardneri*, have been successively reclassified as *Xanthomonas cynarae* pv. *gardneri* and then as *Xanthomonas hortorum* pv. *gardneri* [16,17]. These new data corroborated earlier findings from partial sequencing of housekeeping genes that had suggested close phylogenetic relationships between *X. euvesicatoria* and *X. perforans* and between *X*. *gardneri* and *X. cynarae* [18,19,20,21].

Following this series of taxonomic reclassifications, bacterial spot of pepper and tomato is at present considered to be caused by three different species: *X. euvesicatoria*, *X. vesicatoria*, and *X. hortorum.* While *X. euvesicatoria* pv. *euvesicatoria* and *X. hortorum* pv. *gardneri* strains are well known to be pathogenic of both tomato and pepper plants, *X. vesicatoria* primarily infects tomato plants. *X. euvesicatoria* pv. *perforans* was considered to be restricted to tomato plants, until, in 2012, a strain was isolated from a pepper plant [22]. Historically, the two pathogens *X. euvesicatoria* pv. *euvesicatoria* and *X. vesicatoria* have had a worldwide distribution and were considered as the dominant bacterial spot lineages. More recently, however, *X. euvesicatoria* pv. *perforans* and *X. hortorum* pv. *gardneri* strains are increasingly often isolated in North and South America, Middle East, East Africa and regions bordering the Indian Ocean [23,24,25,26,27,28,29,30,31,32]. Perhaps this wide distribution in different geographical regions is due to introduction of contaminated seeds and/or seedlings [27,29,33,34]. Further, emergence of new strains or lineages through multiple recombination events and acquisition of novel transcription activation-like effector (TALE) may also contribute to this increasing distribution [35,36].

The European and Mediterranean Plant Protection Organization (EPPO) considers all bacterial spot pathogens of pepper and tomato as A2 quarantine pests, which means that they are locally present in the EPPO region [37,38]. For instance, the pathogens have been reported for Italy [39,40], for the Czech and Slovak Republics [41,42] and for the Balkan Peninsula [43]. In the East Balkan, bacterial spot of tomato and pepper was first reported in the 20th century, first on tomato in 1936 (Bulgaria) and then on pepper in 1965 for Bulgaria and in 1999 for North Macedonia [44,45,46]. Since then, the disease has become one of the economically most important diseases of pepper and tomato plants with losses reaching 10% to 20% [47,48]. *X. euvesicatoria* was identified as the dominant species of bacterial spot of pepper plants in Bulgaria and North Macedonia [49,50]. After 2014, the species was also reported as a bacterial spot agent of tomato in Bulgaria [51]. Analyses using restriction fragment length polymorphisms (RFLP), randomly amplified polymorphic DNA (RAPD) markers and rep-PCR revealed substantial genetic diversity among the Bulgarian and North Macedonian *X. euvesicatoria* strains [52,53]. Nevertheless, little information is at present available on the pathogen population structure in Bulgaria and North Macedonia.

Sustainable control measures, such as the use of chemicals, antagonists and/or resistant varieties, will critically depend on a profound knowledge of the population structure and dynamics of the pathogen. For that aim, bacterial typing techniques are nowadays used for reliable and quick differentiation of closely related strains. Over the past decades, a variety of different typing methods have been developed to generate strain-specific patterns. However, only a limited number of molecular tools are able to distinguish strains at high resolution for epidemiological purposes. Whereas multilocus sequence analyses (MLSA) unambiguously identified bacterial spot agents at the species level, they did not emerge as a tool for differentiation of strains within a species due to very few sequence polymorphisms and signs of recombination events in some of the used housekeeping genes [21,27,29,54]. MLSA schemes targeting four to seven housekeeping genes may thus not allow drawing conclusions with respect to the global movement of the pathogens, and are certainly not useful for studying the population structure at a smaller geographical scale (within country) or characterize outbreak situations.

Multilocus variable numbers of tandem repeat analyses (MLVA) have become increasingly popular for high-resolution molecular typing of bacteria because of the high discriminatory power and reproducibility, ease of performance and portability, rapidity and low costs. This is correlated to the greater availability of bacterial genomes, which facilitated their development. Detection and analysis of polymorphic short sequence repeats organized at distinct loci showing high mutation rates has been proven to be a promising tool for epidemiological studies of monomorphic plant-pathogenic bacteria, such as *Pseudomonas syringae, Ralstonia solanacearum*, and several species of *Xanthomonas* [55,56,57,58,59,60,61,62,63,64,65].

Tandem repeat (TR) loci with small (<10 bp) repeat unit sizes are identified as microsatellites. Microsatellites alleles are usually defined from different sizes of DNA fragments revealed after electrophoresis. These electromorphs are then easily scored as repeat numbers, whose variation contributes to discriminate between individuals or populations. The numbers of repeats change through mutational events producing a variable number of tandem repeats (VNTR). These events are favoured by polymerase slippage at DNA replication corresponding to the addition or deletion of repeat motifs [66,67]. Thus, two alleles can share the same electromorph, i.e., are identical in state, following convergent mutational events, without originating from the same ancestral allele, i.e., are not identical by descent. This size homoplasy, where alleles with identical sizes have different evolutionary histories, is related to the process of mutation. Two main models describe the process of mutations at TR loci. Under the stepwise mutation model (SMM), loss or gain of a single TR occurs with equal probabilities. In contrast, under the infinite-allele model (IAM), a new allelic state results from a unique mutational event involving any number of TRs, thereby not counting the actual number of gained or lost repeats [67]. A wealth of data suggested that most microsatellites evolve under an SMM [68]. Another type of variation can be generated within these microsatellite loci due to sequence variation occurring either in the tandem repeats or in the regions flanking the repeats. DNA sequence information can uncover cases of size homoplasy that arise for instance from different combinations of different repeats within a compound microsatellite producing the same electromorph or from small mutations (InDels) within the flanking regions [67,69,70]. This detectable fraction corresponds to the molecularly accessible size homoplasy (MASH).

In order to trace back the spread of bacterial spot pathogens within and among countries, we developed the first MLVA scheme for pepper- and tomato-pathogenic strains of *X. euvesicatoria* by directly assessing the repeat number by sequencing TR loci and applied this new tool to analyse a representative set of 88 *X. euvesicatoria* strains that were isolated from pepper plants in Bulgaria and North Macedonia.

## 2. Materials and Methods

### 2.1. Prediction of VNTR Loci and Primer Design

The complete genome sequence of *X. euvesicatoria* pv. *euvesicatoria* (also known as *Xanthomonas campestris* pv. *vesicatoria*) 85-10 (from pepper, GenBank accession number AM039952) and seven draft genome sequences of *X. euvesicatoria* pv. *euvesicatoria* strains, including one North Macedonian strain (83M, from pepper, acc. no. JSZH00000000,), one Bulgarian strain (66b, from pepper, acc. no. JSZG00000000), two Indian strains (LMG 905, from the unknown host plant, acc. no. JTEI00000000; LMG 918, from pepper, acc. no. JTEK00000000), one strain from Ivory Coast (LMG 909, from pepper, acc. no. JTEJ00000000), one strain from Brazil (LMG 933, from pepper, acc. no. JTEL00000000), and one strain from Tonga (LMG 667, from tomato, acc. no. JTEH00000000) [13,71,72] were screened for the presence of candidate VNTR loci using a bioinformatics pipeline, as previously described (http://www.biopred.net/VNTR/; accessed on 27 November 2020) [58]. In addition, draft genome sequences of representative strains of four other pathovars of *X. euvesicatoria*, all from the so-called Rademaker group 9.2 [73,74], were included: pv. *perforans* strain 91-118 (also known as *Xanthomonas perforans;* from tomato, acc. no. AEQW00000000), pv. *alfalfae* strain CFBP 3836 (also known as *Xanthomonas axonopodis* pv. *alfalfa*; from lucerne, acc. no. AUWN00000000), pv. *allii* strain CFBP 6369 (also known as *X. axonopodis* pv. *allii*; from onion, acc. no. JOJQ00000000), and pv. *citrumelo* strain F1 (also known as *X. axonopodis* pv. *citrumelo*; from citrus, acc. no. CP002914) [75,76,77,78].

For VNTR prediction, parameters were set as follows using the algorithm Tandem Repeats Finder (TRF) [79]: region length, 30 to 1000 bp; unit length, 5 to 9 bp; at least 6 copies and at least 80% similarity between adjacent repeats. Predicted VNTR loci were grouped by homology based on conservation of their 500-bp flanking sequences. Loci were named in the order they were found by the prediction pipeline. For primer design, the homologous flanking regions of the predicted VNTR loci were extracted from all twelve genome sequences and aligned using MUSCLE (http://www.ebi.ac.uk/Tools/msa/muscle/; accessed on 27 November 2020) [80]. PCR primers were designed on the conserved regions and tested for the optimal annealing temperature and dimer formation (http://www.thermoscientificbio.com/webtools/multipleprimer/; accessed on 27 November 2020). Specificity of PCR primers and inter-strain size polymorphisms of PCR amplicons were evaluated by in silico PCR (http://insilico.ehu.es/PCR/; accessed on 27 November 2020).

### 2.2. Bacterial Strains Sampling and Isolation

The bacterial strains used in this study are listed in Appendix A. In order to establish a representative strain collection, a comprehensive survey of pepper plant samples with symptoms of bacterial spot was conducted (Figure 1). Four agro-ecological zones in Bulgaria and three zones in North Macedonia were targeted in this study. Sampling was conducted in diagonal transects in three to four fields at each location. Different parts of affected plants (leaves, stems, flowers, petioles) were collected. For isolation of pathogenic bacteria, 1-cm leaf disks from infected and from healthy plant tissue were cut, disinfected by soaking in a solution of sodium hypochlorite at 2% for 3 min and washed several times with sterile distilled water. Serial ten-fold dilutions to 10^−4^ were performed in sterile physiological salt solution and aliquots of 100 μL suspensions were plated onto King’s B medium plates. Agar plates were incubated at 28 °C for 48 h. Yellow, mucoid, non-fluorescent colonies typical for *X. euvesicatoria* were observed on all plates coming from symptomatic tissue.

### 2.3. Molecular Biological Techniques

A single colony grown on potato-sucrose agar (potatoes 200 g/L, sucrose 20 g/L, and agar 20 g/L) was inoculated in 50 mL lysogenic broth and incubated overnight at 28 °C. The cell density of the bacterial suspension was adapted to an optical density at 600 nm (OD_600nm_) of 1. DNA was extracted from cultivated bacterial cells using the STS kit (STS Ltd., Sofia, Bulgaria) according to the manufacturer’s instructions. The yield and purity of the obtained DNA were monitored on a Nanodrop 2000 spectrophotometer (Thermo Scientific).

PCR amplifications of VNTR loci were performed using genomic DNA of *X. euvesicatoria* strains, including three sequenced reference strains as control: 85-10, 66b, and 83M. PCR reactions were carried out in a final volume of 25 µL and contained 5 µL 5x green GoTaq^®^ reaction buffer, 3 µL 25 mM MgCl_2_, 1 µL oligonucleotide primers at 100 µM, 0.5 µL 10mM dNTPs, and 0.05 µL GoTaq^®^ DNA Polymerase (Promega Corp., Madison, WI, USA). Amplification reactions started with an initial denaturation step at 95 °C for 3 min, followed by 35 cycles each consisting of 30 s at 95 °C, 20 s at 55–58 °C (depending on the primer pair, Appendix A), and 60 s at 72 °C, and finished by an elongation step of 10 min at 72 °C. PCR-amplified VNTR loci were electrophoretically separated on 1% agarose gels to check for specificity of DNA amplification and sequenced (Beckman Coulter Genomics, UK), using one of the PCR primers (Appendix A).

### 2.4. Multilocus Sequence Analysis

Four previously used housekeeping genes were targeted for MLSA: *dnaA*, *fyuA*, *gyrB*, and *rpoD* [20]. DNA amplification was performed as described, and PCR amplicons were sequenced using the forward primers. Sequences, not including the primer sequences, were trimmed to 796 bp (*dnaK*), 628 bp (*gyrB*), 840 bp (*fyuA*), and 846 bp (*rpoD*), totalling to 3110 bp.

### 2.5. VNTR Analysis and Statistics

DNA sequences were aligned using MUSCLE [80] and numbers of complete repeats were derived from multiple sequence alignments. The number of repeats at each locus for every strain was recorded in a matrix and deposited at MLVAbank (http://www.biopred.net/mlva/; accessed on 27 November 2020) [62]. Two different datasets were produced. VNTR-rep data represent the alleles scored as repeat numbers whatever the sequence of the repeat is whereas the VNTR-seq dataset separates alleles according to the number of repeats and the sequence of the VNTR locus. The discriminatory power of each locus was evaluated by the Hunter–Gaston Discriminatory Index (HGDI), which was calculated at http://insilico.ehu.es/mini_tools/discriminatory_power/ (accessed on 27 November 2020).

The evolutionary relationships among the bacterial strains were displayed as minimum-spanning trees using the software PHYLOViZ version 2.0 [81]. Categorical minimum spanning trees were built using the algorithm recommended for TR data, combining global optimal eBURST (goeBURST) and Euclidean distances [82]. Different estimates of genetic diversity as Nei’s index of gene diversity and clonal diversity (Simpson index) were calculated with the poppr 2.2.1 package in R [83]. Allelic richness (A) and private allelic richness (Ap) were computed using the rarefaction procedure for unequal sample sizes with HP-RARE version 1.0 [84]. Nei’s index of gene diversity and analysis of molecular variance (AMOVA) were performed using the Arlequin version 3.5 software package [85]. Levels of significance were determined by computing 999 random permutations. Population pairwise F_ST_ and R_ST_ using Arlequin were computed for the VNTR-rep dataset and their significance was tested using 999 permutations. The population pairwise F_ST_ was only computed for the VNTR-seq dataset as alleles are scored as sequence types, which are not expected to evolve under a stepwise mutation model for which R_ST_ is designed.

## 3. Results

### 3.1. Multilocus Sequence Analysis of Bulgarian and North Macedonian X. euvesicatoria Strains

MLSA has been used to determine phylogenetic relationships among bacterial spot agents of pepper and tomato [21,27,86,87]. All these MLSA schemes, and also those that have been developed for the genus *Xanthomonas*, included a portion of the *gyrB* gene [20,88]. We therefore first evaluated the utility of *gyrB* to differentiate *X. euvesicatoria* from the Balkan Peninsula. Based on previous analyses [52,53], three strains from North Macedonia (M) and nine strains from Bulgaria (b) were chosen: 1M, 5M, 7M, 29b, 30b, 38b, 54b, 61b, 67b, 74b, 82b, and 86b. Sequence comparison of a 628-bp DNA fragment revealed that all twelve sequences were identical to each other and to the corresponding sequences from the eight completely sequenced strains of *X. euvesicatoria* pv. *euvesicatoria*, 85-10 (Florida), 83M (North Macedonia), 66b (Bulgaria), and five LMG strains (667, 905, 909, 918, 933) from four continents (India, Ivory Coast, Brazil, Tonga) [13,71,72].

To obtain more sequence information, we further sequenced a portion of the three remaining genes of Young’s MLSA scheme, *dnaK*, *fyuA*, and *rpoD* [20]. Again, all DNA sequences from the strains 7M, 29b, 38b, 67b, and 86b, totalling to 3110 bp (including *gyrB*), were identical to each other and to the sequences from the eight reference strains. These data confirm that the *X. euvesicatoria* pepper pathogens are largely monomorphic and that MLSA is not a useful tool for epidemiological studies at any geographical scale.

### 3.2. Identification of Polymorphic VNTR Loci and Development of a 16-Loci MLVA Scheme (MLVA-16)

The screening of twelve available genome sequences identified 76 VNTR loci with 5-bp to 9-bp repeats. Based on conserved presence of the loci and observed polymorphisms, 28 promising loci were selected to develop an MLVA scheme. Candidate VNTR loci were first analysed on a test panel of 15 strains representing worldwide diversity, thus assessing the conservation of the loci (Appendix A). Six Bulgarian strains (24b, 27b, 28b, 73b, 89b, and 90b) collected from five different pepper cultivars in two different regions and four North Macedonian strains (54M, 65M, 66M, and 80M) from two regions and two cultivars were chosen for this purpose. Two pairs of strains isolated from the same cultivar in the same region and in the same year (65M, 66M from North Macedonia and 89b, 90b from Bulgaria) were included to exemplify the resolutive power of the VNTR loci. In addition to the ten strains from the Balkan Peninsula, we included five strains from four more continents. Upon PCR amplification and confirmation by agarose gel electrophoresis, 16 markers were selected for further analyses. All 16 primer pairs led to the amplification of a single dominant DNA fragment, which allowed sequencing using one of the two PCR primers.

Mapping of the 16 loci to the manually annotated genome sequence of strain 85-10, which contains four plasmids, revealed that all of them are encoded on the chromosome without any obvious clustering (Table 1). The majority of repeat arrays are found in intergenic regions. Only two loci, Xe_03 and Xe_17, are present in coding sequences. Since both loci consist of 6-bp repeat arrays, repeat number variation does not affect the reading frame. Locus Xe_17 is found in gene XCV3092, which is predicted to encode an NADPH-dependent sulphite reductase. The repeat array encodes a string of Ala-Asp motifs ([AD]_8_ in strain 85-10), which separate the *N*-terminal flavodoxin domain from the C-terminal oxidoreductase NAD-binding domain. The second locus, Xe_03, that is found in a coding sequence belongs to the *xopD* type III effector gene, XCV0437, in strain 85-10. Here, the repeat array codes for the motif (Lys-Ala)_3_-(Glu-Ala)_3_-(Lys-Ala)_3_-(Glu-Ala-Lys-Ala)_2_. This motif is found at the end of the so-called *N*-terminal extension, which had been overlooked during manual annotation but later shown to be translated [89].

### 3.3. Application of the MLVA-16 Scheme on a Collection of Pepper-Pathogenic X. euvesicatoria from Bulgaria and North Macedonia

Sixteen VNTR loci were used to study a collection of 88 *X. euvesicatoria* strains from Bulgaria and North Macedonia (Appendix A). PCR-amplified DNA fragments were sequenced using one of the PCR primers. Numbers of complete repeats were then derived from multiple sequence alignments. The number of alleles per locus ranged from two to eight for loci Xe_29 and Xe_04, respectively (Table 1). The number of alleles covered the full allelic range for nine loci out of 16 (Table 1). On average, 4.5 alleles were observed per locus. The discriminant power of the 16 loci, as estimated by HGDI (Hunter–Gaston discriminatory index) scores, ranged from 0.190 (Xe_03, Xe_22, and Xe_34) to 0.803 (Xe_09). Six loci had poor discriminatory power (HGDI score < 0.3) for the set of *X. euvesicatoria* strains from the Balkan Peninsula (Xe_02, Xe_03, Xe_07, Xe_16, Xe_22, Xe_34). Twenty-five per cent of the VNTR loci had high discriminatory power (HGDI score > 0.6) (Xe_04, Xe_09, Xe_14, Xe_15).

Combining all 16 loci into an MLVA-16 scheme resulted in an HGDI score of 0.944 for the 88 Balkan strains and allowed to resolve 36 MLVA haplotypes (MTs), thus demonstrating their suitability for typing pepper-pathogenic *X. euvesicatoria* strains. The 36 MTs grouped in eight clonal complexes, i.e., groups of single-locus variants (SLVs), representing 26 MTs and 68 strains (Figure 2). Among these SLVs, 73.7% were single-repeat variants (SRVs) and 21.1% differed by two repeats, which might result from a sampling bias where an evolutionary step is missing. Among the nine loci showing SLVs, seven exhibited exclusively SRVs, Xe_49 produced a double-locus variant and Xe_04 evolved with variations involving from one to three repeats.

### 3.4. Impact of Homoplasy on VNTR Typing

Since individual repeats within a VNTR locus often vary slightly in sequence, DNA sequencing provides a tool to discover cases of size homoplasy. This fraction of homoplasious loci is called molecularly accessible size homoplasy [67]. Indeed, only six loci (Xe_02, Xe_04, Xe_06, Xe_14, Xe_29, and Xe_49) were perfect multiple repeats which consisted of a single repeat type, whereas the other loci were composed of two to four repeat types (Table 1). Almost all the different repeats within composite VNTR loci differed by only one base pair except for locus Xe_15, where one repeat differed from the other two by two base pairs. However, 12 of the 17 alternative repeat types were found at the end of a repeat array, and only once per repeat array. Only five loci, Xe_03, Xe_09, Xe_11, Xe_15 and Xe_34, were really chimeric loci consisting of several alternate repeat types along the entire array.

Among these composite VNTR loci, three (Xe_11, Xe_15 and Xe_34) showed size homoplasy where the same allele, i.e., number of repeats, could result from different combinations of repeat types or sequence type ST (Table 2). For instance, upon DNA sequencing we noted that locus Xe_11 consists of three different 7-bp repeat motifs, from which two different combinations produced PCR amplicons with the same size, corresponding to 13 repeats (Table 2). We designated the two homoplasious alleles as Xe_11-ST1(13) and Xe_11-ST2(13). While these two alleles could be resolved by DNA sequencing they cannot be resolved by other techniques (e.g., capillary electrophoresis) that only deliver amplicon sizes.

Resolving cases of homoplasy slightly increased the discriminatory power of the three affected loci (Table 1). Yet, correcting for homoplasy did not change the number of MLVA haplotypes and likewise did not increase the clonal diversity estimated by the Simpson index (Table 3).

### 3.5. Single-Nucleotide Polymorphisms in the Flanking Sequences

DNA sequencing did not only reveal sequences of the tandem repeats but also provided insight into the sequence conservation of the flanking regions of the repeat arrays. Similar to the four housekeeping genes, DNA sequences were highly conserved. Yet, two VNTR loci showed single nucleotide polymorphisms (SNPs) in their flanking sequences. For ten strains (13b, 42b, 43b, 62b, 67b, 73b, 74b, 96b, 102b, 35M), we observed a C to G exchange 26 base pairs upstream of Xe_09 within the coding sequence of gene XCV0924, leading to a silent mutation (threonine codon ACC –> ACG). Two additional SNPs were shared among the same ten strains in the intergenic region between *fliP* (XCV1988 in strain 85-10) and *fliQ* (XCV1987 in strain 85-10), 4 and 119 bp upstream of Xe_14. Hence, information from the conserved flanking regions divided the set of strains into two genetic clusters (GC), one consisting of ten strains (nine from Bulgaria and one from North Macedonia) sharing three SNPs, which comprise seven MTs (1, 2, 15, 16, 17, 18, 19) (GC2), and another cluster comprising the rest of the strains (GC1; Appendix A).

Interestingly, all ten strains of GC2 shared the same flanking sequences of Xe_09 and Xe_14 with strains 85-10 (Florida), 66b (Bulgaria), and four LMG strains (667, 905, 909, 933 from New Zealand, India, Ivory Coast, Brazil), collectively originating from six continents, but not with strain 83M from North Macedonia, which belongs to GC1 (Appendix A) [13,71,72]. This finding suggests that strains of GC2 are phylogenetically linked to a worldwide lineage.

### 3.6. Population Structure of X. euvesicatoria

The population of *X. euvesicatoria* from Bulgaria showed a slightly greater genetic diversity than the North Macedonian population, as shown by the expected number of multilocus genotypes (eMLG = 21.5 and 18.0, respectively) and the allelic richness, with A = 4.25 and 3.13, respectively (Table 3). The private allelic richness was much greater for the Bulgarian population (Ap = 1.25) than for the North Macedonian strains (Ap = 0.13). These estimates of genetic diversity based on allelic data were a little higher when estimated from the VNTR-seq dataset (Table 3, numbers in brackets) but those estimated from haplotypes did not change.

The Bulgarian and North Macedonian populations were clearly differentiated when computed from both the VNTR-rep dataset (F_ST_ = 0.101, *p* < 0.001/R_ST_ = 0.108, *p* = 0.002) and from the sequence-corrected VNTR-seq dataset (with F_ST_ = 0.098, *p* < 0.001).

Analysis of molecular variance (AMOVA) partitioning different regional levels revealed that most of the genetic variation was significantly explained by the variation between regions within countries and by the intra-region variations. In contrast, inter-country variation did not significantly contribute to the total genetic variation (1.05%, *p* = 0.47) (Table 4). Most of the pairwise comparisons of the regional populations were significantly differentiated (Table 5). However, none of the pairwise R_ST_ parameters between Bulgarian regions B3 and B4, the North Macedonian region M1 and the North Macedonian region M2 bordering Bulgaria were significantly different (Table 5).

The haplotypic diversity estimates (eMLG and Simpson index) of Bulgarian and North Macedonian regional populations were in the same range (Appendix A). However, the allelic diversity estimates were slightly greater for the Bulgarian regional populations, which had also more private alleles than the North Macedonian regions.

### 3.7. Strains from Bulgaria and North Macedonia Are Genetically Closely Related

The relationship among the strains based on the MLVA-16 results is presented in a minimum-spanning tree (Figure 2). The minimum spanning tree produced from the VNTR-rep dataset grouped the 36 MLVA haplotypes (MTs) in eight small clonal complexes (68 strains) and ten singletons (20 strains), i.e., haplotypes that differed by more than one locus from all other haplotypes. The clonal complexes are formed by two (CC8) to four (CC1, CC2, CC3) haplotypes. Five clonal complexes (CC1, CC3, CC4, CC7, CC8) grouped strains from both countries, two other clonal complexes (CC5 and CC 6) grouped strains from different regions of Bulgaria, and the last clonal complex (CC2) grouped only strains from the Western region of North Macedonia (Figure 2). A sub-group of nine MTs including CC3, CC5 and MTs 5 and 10 are double-locus variants, perhaps due to a sampling bias with an evolutionary step missing, and could thus be epidemiologically related.

Strains from the different regions of Bulgaria and North Macedonia were found to be widespread in genetically distant clonal complexes or singletons. Several strains of different regions, except for B1, shared haplotypes with strains from other regions or the other country (Figure 2). The large geographical distribution of haplotypes and the existence of epidemiologically closely related strains in different regions and countries suggest long dispersal of strains.

Most of the haplotypes of the network differed by five loci or less and grouped as a large genetic cluster, corresponding to GC1. This cluster was well separated from a phylogenetically distant second cluster (GC2) composed of two clonal complexes (CC6, CC7) and the singleton MT15 by variations at 12 loci or more. Notably, cluster GC2 contained mostly strains from three Bulgarian regions and one strain from North Macedonia.

Interestingly, only the GC2 strains showed the SNPs detected in the flanking regions of loci Xe_09 and Xe_14 (see before), in comparison to GC1 strains. In addition to these two polymorphic loci, specific alleles of GC2 were observed for eight loci (Xe_03, Xe_07, Xe_10, Xe_11, Xe_15, Xe_17, Xe_22, Xe_34). Among them, Xe_34, which was monomorphic in GC1 strains sharing a unique allele of four repeats, had different haplotypes (12 to 15 repeats) and two homoplasious alleles with 13 repeats in GC2 strains. Similarly, GC2 strains had specific sequence types for two additional homoplasious alleles, i.e., ST1 for Xe_11 (all strains) and ST2 for the Xe_15 allele with nine repeats (CC6) (Table 2).

The relationships among the haplotypes in the minimum spanning tree obtained from sequence-corrected data (VNTR-seq dataset) did not change much except that cluster GC2 strains were placed as a 13-loci variant of MT8 (CC4) instead of a 12-loci variant of MT10 (Appendix A). However, considering the large distance between GC1 and GC2 it is not very meaningful to connect them at all. Rather, our data suggested that our sample consists of two groups of strains with one of them (GC2) being linked to a worldwide pandemic expansion and another one (GC1) being restricted to the Balkan Peninsula at the time being.

### 3.8. Transferability of VNTR Markers to Other Pathovars of X. euvesicatoria

Here, we used genomic information for strains from different pathovars of *X. euvesicatoria*, including *perforans* (strain 91–118), *alfalfae* (strain CFBP 3836), *allii* (strain CFBP 6369) and *citrumelo* (strain F1), for prediction of VNTRs and primer design. This way, we expected to develop a scheme that can be used beyond the pathovar *euvesicatoria*. To test this hypothesis, we extracted the 16 VNTR loci from all available genome sequences of *X. euvesicatoria* and analysed them in silico (Table 6). Except for Xe_03, 15 out of the 16 loci were present in all the other pathovars (*alfalfae*, *allii*, *citrumelo*, *commiphorea*, *dieffenbachiae*, *perforans*). Interestingly, pathovar *perforans*, which causes the same disease as the pathovar *euvesicatoria* but was ranked as a different species when we started the project, shared all 16 loci with pathovar *euvesicatoria*. Remarkably, most loci in pathovar *perforans* were polymorphic and only two of them, Xe_06 and Xe_07, appeared to be monomorphic. Compared with pathovar *euvesicatoria*, allelic ranges in *perforans* appeared to be smaller, but this observation may be due to sampling biases, as most *perforans* strains were from a restricted geographic area.

## 4. Discussion

Here, we developed and applied the first VNTR scheme for the pepper- and tomato-pathogenic strains of *X. euvesicatoria*. VNTR typing has many advantages in comparison with other techniques with a similar level of taxonomic resolution, such as amplified fragment length polymorphism (AFLP), randomly amplified polymorphic DNA (RAPD) and repetitive sequence-based rep-PCR [90]. Whereas traditional fingerprinting techniques generate more or less complex profiles, which are often technically demanding and hard—if not impossible—to compare between laboratories, VNTRs deliver integer numbers of repeats, which can be easily electronically stored and compared. VNTRs represent single-locus markers that are extremely informative and can be analyzed as allele frequencies for population genetics and for effectively tracing bacterial outbreaks or dispersal. This is also the reason why MLVA schemes, i.e., VNTR schemes targeting between 8 and >20 molecular markers, have become more and more popular in the field of plant pathology. Strikingly, MLVA schemes are now available for seven economically important species of *Xanthomonas*, namely *X. arboricola* (pvs. *corylina*/common hazel, *fragariae*/strawberry, *juglandis*/Persian walnut, *populi*/poplar and other trees, *pruni*/stone fruits like almond, apricot, cherry, peach and plum), *X. citri* (pvs. citri/citrus, *mangiferaeindicae*/mango, *viticola*/grapevine), *X. euvesicatoria* (pv. *euvesicatoria*/pepper and tomato), *X. fragariae* (strawberry)], *X. oryzae* (pvs. *oryzae* and *oryzicola*)/rice), *X. phaseoli* (pv. *manihotis*/cassava) and *X. vasicola* (pv. *musacearum*/banana) [55,56,58,60,61,62,63,64,65,91,92,93,94,95]. In order to allow and stimulate comparisons with other datasets, we deposited our data at MLVAbank (http://www.biopred.net/MLVA/, accessed on 27 November 2020).

Whereas existing MLSA schemes target all members of the *Xanthomonas* genus, they suffer from a comparatively low level of taxonomic resolution and cannot efficiently resolve strains within a pathovar [20,92,96]. On the other hand, high-resolution MLVA schemes targeting small repeats, i.e., microsatellites, are typically developed for a distinct pathovar and are limited in their applicability to other pathovars of the same species. For instance, only a subset of the VNTR loci that were employed for an MLVA scheme of *X. oryzae* pv. *oryzae* could also be used for the other pathovar, *oryzicola* [58,62]. Similarly, starting with a set of 26 VNTR loci for the species *X. arboricola*, only 9 to 23 of them were useful to characterize strains from five different pathovars [60]. Yet, in some cases, schemes developed for a distinct pathovar proved to be useful for other pathovars of the same species [55,56]. Based on available genome sequences we estimated the transferability of our VNTR markers to other pathovars of *X. euvesicatoria*. In total, 14 of the 16 VNTR markers were found to be polymorphic in the pathovar *perforans* and thus useful for molecular typing. We are not sure how valuable, i.e., polymorphic, the loci are for the other pathovars because we had only access to very few genome sequences for them. However, at least for those VNTRs with moderate or large repeat numbers (i.e., four or more), we expect size polymorphisms for most of them [97].

Size homoplasy originating from different combinations of repeats was detected for five haplotypes in three of 16 VNTR loci (Xe_11, Xe_15, Xe_34; Table 2), as revealed by DNA sequencing. In these cases, haplotypes with the same number of repeats resulted from different combinations of repeat types in these loci. Although size homoplasy at three VNTR loci somewhat reduced the genetic diversity score, this effect did not have a major impact on the population structure and phylogenetic relationships among the analyzed strains. Sequence variation was also observed in the flanking regions of two loci Xe_09 and Xe_14 but they did not lead to size homoplasy. Remarkably, sequence types were lineage-dependent, thus supporting the population structure with two major lineages. Moreover, size homoplasy did not occur within clonal complexes but between genetically remote strains that could originate from different populations as shown previously for different populations of *X. citri* pv. *citri* [93]. Therefore, the information from the sequences (VNTR-seq datasets) strongly supports the conclusion drawn from alleles scored as repeat numbers (VNTR-rep datasets).

While DNA sequencing allowed to decipher the relatively small impact of homoplasy on typing, it may not be advantageous and cost-saving to use this approach for routine typing. However, this MLVA-16 scheme could easily be used in a high-throughput typing technique based on capillary sequencer technology and a multiplexing strategy [96]. Primer pooling is based on annealing temperature (Appendix A), expected amplicon size and the use of different fluorescent dyes. We propose to multiplex four primer pairs, each with a specific dye (Appendix A), as previously reported for other xanthomonads [62,65,95].

The sequence polymorphisms in the flanking regions of two VNTR loci (Xe_09, Xe_14), i.e., three SNPs, divided the sample into two major lineages, GC1 and GC2, which correlated with the gross population structure based on the MLVA haplotypes. We therefore conclude that the minimum spanning tree with its two major branches is robust because this topology is supported by the three SNPs, which are likely to evolve slower. Interestingly, SNPs shared among GC2 strains were also observed in the genome sequences of other strains originating from six continents (Europe, North America, South America, Africa, Asia, Oceania). This finding suggests that strains of this group are phylogenetically linked to a worldwide lineage and prompted us to check if sequence types that appeared to be restricted to the Balkan Peninsula have been found elsewhere. BLASTN analysis of whole-genome sequences of pepper/tomato pathogens at NCBI GenBank revealed that 46 *X. euvesicatoria* strains share the three SNPs with GC1 strains whereas only five strains contain the three SNPs of GC2 strains, among them 83M from North Macedonia, strains 259, 315 and 329 from the United States of America, and strain BRIP62438 from Australia. Three strains at GenBank did not share any of the sequence types at the flanking region of Xe_14, among them an atypical strain (LMG 918), which does not cluster with any of the two pepper/tomato-pathogenic pathovars of *X. euvesicatoria* (pvs. *euvesicatoria* or *perforans*) [13,98], strain NI38, which is another atypical strain originating from Nigeria [86], and strain LMG 27970, for which the assembled genome sequence with its 1491 contigs lacks this locus [14].

Molecular epidemiology is based on our ability to differentiate the individuals and to estimate their genetic relatedness to decipher the possible propagative pathways and the origin of the inoculum. Both the high discriminatory power of the VNTR markers and the sequence variation within these loci supported the existence of two major lineages of *X. euvesicatoria* pv. *euvesicatoria*, one of which appears to be overrepresented on the Balkan Peninsula. The smaller GC2 corresponds to worldwide distributed SNP haplotypes in the flanking region of two VNTR loci, suggesting introduction into Bulgaria and North Macedonia from a dominant worldwide lineage, whereas the majority of strains (GC1), distributed in all the sampled regions of Bulgaria and North Macedonia, are endemic. Our data support at least three introduction events at the origin of the Balkan population of *X. euvesicatoria*. The two clonal complexes of GC2 are phylogenetically remote and could originate from two different introduction events. GC1 represents a group of more or less genetically related strains whose ancestor was probably introduced during an older event. Our data further indicate a strong genetic link between the North Macedonian and Bulgarian populations of *X. euvesicatoria*. A part of the North Macedonian population probably originated from Bulgaria, consecutively to one or few introduction events. This conclusion is supported by the date of disease reports, greater genetic diversity and private allelic richness observed in Bulgaria, the absence of genetic differentiation between the North Macedonian regions bordering Bulgaria and some Bulgarian regions and the sharing of haplotypes or clonal complexes by strains from those regions.

Long-distance dispersal through infected plant material (seeds) most probably occurred within and between countries. Seeds can serve as an important source of inoculum and this could favor the introduction of new strains [4]. More analyses of strains from various regions in the world are required to definitively disclose the origin of these two lineages of *X. euvesicatoria* in the Balkan area. Our MLVA tool together with the informative SNPs will be instrumental in such analyses and will contribute to a better understanding of dispersal pathways and the role of infected seeds in the epidemiology of *X. euvesicatoria*.

## Figures and Tables

**Figure 1 microorganisms-09-00536-f001:**
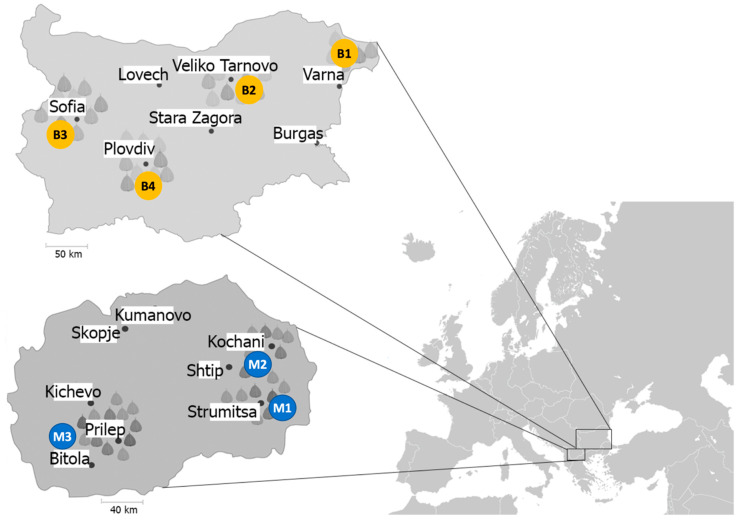
Geographic map depicting Europe with a close-up of Bulgaria (top) and North Macedonia (bottom) indicating the origin of the analysed strains, corresponding to four Bulgarian regions (B1 to B4) and three North Macedonian regions (M1 to M3).

**Figure 2 microorganisms-09-00536-f002:**
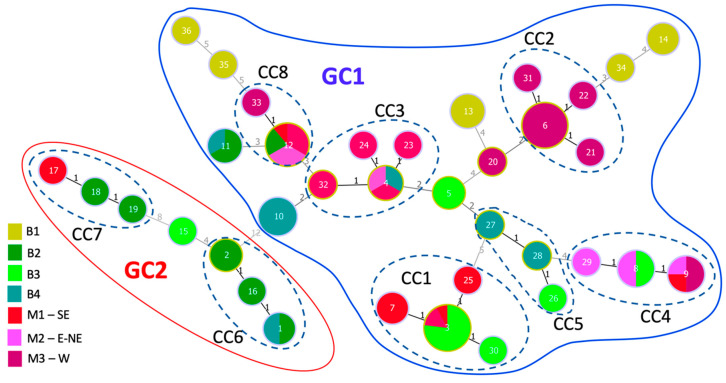
Categorical minimum spanning tree from the VNTR-rep dataset using the goeBURST and Euclidian algorithms. Numbers in circles identify the 36 MLVA haplotypes. The circle sizes are proportional to the number of strains per haplotype, with the smallest circles (e.g., no. 17 in CC7) corresponding to one strain and the largest circle (no. 3 in CC1) corresponding to 13 strains. Colours indicate the country and region of origin, with four Bulgarian regions (B1 to B4) and three North Macedonian regions (SE, South East; E-NE, East-North East; W, West). Black numbers between the circles connect single-locus variants (SLVs) grouped into eight clonal complexes (CC), which are encircled with dashed lines. Gray numbers indicate the number of loci (>1) that differ between connected MLVA haplotypes. Two major genetic clusters, as defined be specific SNPs (GC1 and GC2), are encircled by solid blue (GC1) and red (GC2) lines (see Section 3.5).

**Table 1 microorganisms-09-00536-t001:** Characteristics of variable number of tandem repeats (VNTR) and Hunter–Gaston Discriminatory Index (HGDI) scores for the 16 loci of the multilocus variable number of tandem repeat analyses (MLVA) scheme.

Locus	Position in Strain 85-10 ^1^	Dominant Repeat Type	Other Repeat Types ^2^	No. of Alleles ^3^	Allelic Range ^4^	HGDI Score ^5^
Xe_02	215122..215170	TCCCCAT	-	4	4–7 #	0.286
Xe_03	487069..487152	TTTGGC	TCTGGC * TTCGGC TTTGTC *	3	12–14 #	0.190
Xe_04	624229..624277	CGATTCC	-	8	5–12#	0.764
Xe_06	857148..857196	AACAGCC	-	3	6–8 #	0.317
Xe_07	924719..924767	CCGGGTC	CCGGGCC *	4	4–7 #	0.211
Xe_09	1053822..1053863	GGGATTT	GGGATTC GGGAATC	7	6–18	0.803
Xe_10	1222069..1222110	AGGCGGT	AGGCGGC *	6	5–12	0.575
Xe_11	1504314..1504418	CCGATTC	CCTAATC CCCAATC	5 (6)	11–16	0.455 (0.480)
Xe_14	2268785..2268850	ACAGCG	-	6	6–11 #	0.738
Xe_15	3198440..3198527	GCAGACAG	GCAGGCAG GCAGAGAT *	5 (8)	6–10 #	0.688 (0.779)
Xe_16	3505639..3505687	AATGGGG	AATCGGG *	3	5–9	0.263
Xe_17	3514941..3514994	TCGGCA	TCGGCG *	5	9–14	0.437
Xe_22	4396287..4396335	TTGGCGG	TTGGCGC *	3	5–10	0.190
Xe_29	4211581..4211608	CGATTCC	-	2	4–5 #	0.315
Xe_34	458055..458096	GATTCGG	GAATCGG GAATTCG * GAATCCG *	4 (5)	5–16	0.190 (0.192)
Xe_49	4410313..4410342	TGGCCG	-	4	5–8#	0.575
MLVA-16				36 haplotypes		0.944

^1^ Genomic coordinates in GenBank accession number AM039952. ^2^ Alternative repeat types that are only found once per repeat array at its end are indicated by a *. ^3^ Number of alleles from the VNTR-rep dataset. Numbers in brackets take resolved cases of homoplasy, i.e., different sequence types, into account (from the VNTR-seq dataset). ^4^ Allelic ranges that contain all alleles are indicated by a #. ^5^ Hunter–Gaston discriminatory index (HGDI) scores of individual VNTR loci and of the 16-loci VNTR analysis, MLVA-16. Numbers in brackets correspond to the VNTR-seq dataset.

**Table 2 microorganisms-09-00536-t002:** VNTR loci with size homoplasy, as revealed by their sequence type (ST).

VNTR Locus	No. of Repeats	ST	Strains ^1^	Repeat Pattern
Xe_11	13	1	CC6 (**62b**, **67b**, **74b**, **96b**, **102b**)/CC7 (**42b**, **43b**, **35M**)/ MT15 (**13b**)	(CCGATTC)_7_-(CCCAATC)_1_-(CCTAATC)_1_-(CCCAATC)_4_
Xe_11	13	2	CC3 (10b, 1M, 2M, 5M, 25M, 50M)/MT10 (44b, 45b, 47b, 49b, 51b)	(CCGATTC)_6_-(CCCAATC)_1_-(CCTAATC)_1_-(CCCAATC)_5_
Xe_15	10	1	CC3 (10b, 1M, 2M, 5M, 25M, 50M)/MT11 (61b, 69b, 70b)	(GCAGGCAG)_3_-(GCAGACAG)_6_-(GCAGAGAT)_1_
Xe_15	10	2	CC8 (105b, 106b, 77M, 79M, 80M, 81M, 82M, 83M, 84M, 86M)/MT35 (80b)	(GCAGGCAG)_2_-(GCAGACAG)_7_-(GCAGAGAT)_1_
Xe_15	9	1	CC1 (5b, 12b, 24b, 27b, 29b, 30b, 31b, 38b, 55b, 56b, 7M, 11M, 28M, 31M, 37M, 38M)/ CC5 (39b, 93b, 94b)/MT5 (11b, 54b)/MT10 (44b, 45b, 47b, 49b, 51b)/MT14 (81b, 82b)/MT34 (85b)/MT36 (86b)	(GCAGGCAG)_2_-(GCAGACAG)_6_-(GCAGAGAT)_1_
Xe_15	9	2	CC6 (**62b**, **67b**, **74b**, **96b**, **102b**)	(GCAGGCAG)_1_-(GCAGACAG)_7_-(GCAGAGAT)_1_
Xe_15	8	1	CC1 (28b)/MT13 (77b, 78b, 79b)	(GCAGGCAG)_2_-(GCAGACAG)_5_-(GCAGAGAT)_1_
Xe_15	8	2	CC2 (54M, 55M, 56M, 57M, 58M, 61M, 62M, 63M,64M, 65M, 66M, 67M, 68M, 69M, 70M)/CC4 (89b, 90b, 71M, 72M, 73M, 74M, 76M, 85M, 87M)/MT20 (59M)	(GCAGGCAG)_3_-(GCAGACAG)_4_-(GCAGAGAT)_1_
Xe_34	13	1	CC7 (**42b**, **35M**)	(GATTCGG)_5_-(GAATCGG)_2_-(GATTCGG)_1_-(GAATCGG)_4_-(GAATTCG)_1_
Xe_34	13	2	CC6 (**62b**, **67b**, **74b**, **96b**, **102b**)	(GATTCGG)_4_-(GAATCGG)_2_-(GATTCGG)_1_-(GAATCGG)_5_-(GAATTCG)_1_

^1^ Strains belonging to cluster GC2 are indicated in bold. Clonal complexes (CC) and singleton MLVA haplotypes (MT) are underlined twice if they contain all strains and once if they contain only a subset of strains.

**Table 3 microorganisms-09-00536-t003:** Genetic diversity parameters of *X. euvesicatoria* from Bulgaria (*n* = 45) and North Macedonia (*n* = 43) estimated from the two MLVA-16 datasets (VNTR-rep and VNTR-seq).

Country	Polymorphic Loci	eMLG ^1^	Simpson Index D	H_E_ ^2^ (seq) ^4^	A ^3^ (seq) ^4^	Ap ^3^ (seq) ^4^
*Both countries*		22.9	0.944	0.437	-	-
Bulgaria	16	21.5	0.932	0.494 (0.506)	4.25 (4.56)	1.25 (1.44)
North Macedonia	15 ^5^	18	0.890	0.329 (0.332)	3.13 (3.25)	0.13 (0.13)

^1^ eMLG, expected number of multilocus genotypes estimated from a rarefaction procedure (*n* = 43). ^2^ H_E_, Nei’s index of gene diversity. ^3^ A, allelic richness; and Ap, private allelic richness estimated from a rarefaction procedure (*n* = 43). ^4^ Numbers are estimated from the VNTR-rep dataset whereas numbers in brackets are from the VNTR-seq dataset, which take resolved cases of homoplasy into account (homoplasy for VNTR loci Xe_11, Xe_15 and Xe_34). ^5^ The VNTR locus Xe_02 is monomorphic for North Macedonian strains.

**Table 4 microorganisms-09-00536-t004:** Analysis of molecular variance (AMOVA) partitioning different geographical levels (regions and countries) estimated from R_ST_ parameters based on the VNTR-rep dataset.

Source of Variation	D.f.	Sum of Squares	Variance Components	Percentage of Variation	*p*-Value
- between countries	1	177.11	0.329	1.05	0.437
- between regions within countries	5	706.15	10.05	32.14	<0.001
- within regions	81	1691.68	20.88	66.80	<0.001
Total	87	2574.94	31.26		

**Table 5 microorganisms-09-00536-t005:** Genetic differentiation estimated by R_ST_ pairwise comparisons of regional collections of *X. euvesicatoria* strains from Bulgaria (B1-B4) and North Macedonia (M1-M3) based on the VNTR-rep dataset.

	B2	B3	B4	M1	M2	M3
B1	0.442 **	0.304 ***	0.239 **	0.250 **	0.321 *	0.501 ***
B2		0.400 **	0.358 **	0.406 **	0.447 *	0.645 ***
B3			0.012 NS	−0.031 NS	0.070 NS	0.313 ***
B4				−0.040 NS	−0.014 NS	0.200 **
M1					−0.001 NS	0.237 ***
M2						0.451 ***

*p* values: NS, non significant, i.e., *p* > 0.05; *, 0.01 < *p* < 0.05; **, *p* < 0.01; ***, *p* < 0.001.

**Table 6 microorganisms-09-00536-t006:** Allelic range of VNTR loci from different pathovars of *Xanthomonas euvesicatoria*.

Locus	*euvesicatoria*	*perforans*	*alfalfae*	*allii*	*citrumelo*	*commiphoreae*	*dieffenbachiae*
Xe_02	2–11	1–16	13	10	12	18	15
Xe_03	8–14	11–12	NA	NA	NA	NA	NA
Xe_04	3–11	2–3	3	9	3–4	4	4
Xe_06	3–11	2	2	4	2	6	2
Xe_07	3–8	4	2	4	2–8	6	6
Xe_09	5–16	5–7 (11)	5-6	12	12–13	6	14
Xe_10	3–8	(3) 5–7	4	4	4	5	10
Xe_11	9–17	4–6	4	9	5–6	9	4
Xe_14	2–13	5–8	5	9	8–9	6	9
Xe_15	6–11	2 (6.5)	2	2	4–6	6	2
Xe_16	3–16	(4) 10–12	12	4	4	4	11
Xe_17	8–18	10–16	9	15	9–12	15	10
Xe_22	5–21	3–9	5	12	4	16	8
Xe_29	3–4	4–7	2–9	5	5	4	3
Xe_34	4–15	9–14	7–12	8	8–11	15	7
Xe_49	7–18	(4) 6–7	6–7	4	9–15	10	9
No. of strains	54	143	2	1	4	1	1

## Data Availability

All data produced in this study are presented in this publication.

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
