# Peer review of "Molecular Epidemiology of Xanthomonas euvesicatoria Strains from the Balkan Peninsula Revealed by a New Multiple-Locus Variable-Number Tandem-Repeat Analysis Scheme"

_microorganisms, 2021, doi:10.3390/microorganisms9030536_

Round 1
Reviewer 1 Report
This is a well written manuscript that implements a new typing scheme for epidemiological studies of Xanthomonas euvesicatoria. Similar MLVA approaches have been valuable for studying other bacterial plant pathogens. The authors provide a detailed analysis of strains from Bulgaria and Macedonia, examine the transferability of their markers to other pathovars using in silico analysis of genomes, and provide a comprehensive introduction to their work. I have relatively minor comments and clarifications.
- Throughout the manuscript the terms “isolates” and “strains” seem to be used interchangeably. Different bacteriologists have different definitions for these terms, so it would be good to define their use here or only use one of them.
- The introduction addresses bacterial spot on both tomato and pepper and the objective is to develop an MLVA scheme for pathogens of both. But very few tomato strains were used (via genomes in databases) and all strains examined from Bulgaria and Macedonia were from pepper. This is fine, but I think it needs to be stated that the populations examined were on pepper in the last sentence of the introduction. Similarly, the first paragraph in the methods gives the geographic origin but not the host origin of strains whose genomes were used to develop markers.
- Are the authors were suggesting that users of the MLVA scheme should sequence each locus because that’s what they did or use them for fragment analysis? Sequencing each locus is not particularly low cost and a discussion of the costs and benefits versus whole genome sequencing is warranted. Later in the manuscript, when the authors discuss VNTR-rep vs VNTR-seq, I realized that this is showing that the scheme will also work by fragment length analysis, but there is no indication given of fragment lengths or guidance for multiplexing the markers, which is common practice for fragment analysis. Clarification on these issues would be helpful for those considering whether to adopt this scheme.
- Line 205: It is awkward that the statement about concentration of the bacterial suspension is given after the sentence about DNA extraction. The sentences should be swapped.
- Line 217: PCR products were run on 1% agarose gel to check for amplification or to screen for a single band?
- I understand that the results for each locus were deposited in an online database, but it would be helpful for readers of the manuscript to have a supplementary table giving the repeats at each locus for each strain.
- Line 328: Is double-repeat variant meant here? I don’t see how a single marker can produce a double-locus variant.
- Figure 1: Please indicate in the legend the difference between the black and gray numbers on the paths. If more than one locus was different between haplotypes, are these numbers the total repeat differences or average? It would be helpful to have a legend for node size showing the circle size for 1 strain and also the haplotype with the largest number of strains.
- It’s not clear to me if the information on homoplasy in Table 3 is factored into the VNTR-rep dataset or only the VNTR-seq dataset. Or it didn’t change the MLVA haplotypes and so is irrelevant?
- I find the SNP results confusing. It would be helpful to have a table or figure that summarizes these results including the strains and MTs grouped by the SNPs. On lines 376-377, are the strains differentiated by SNPs also in distinct MTs? The MTs are listed but it is not clear if the strains with the alternate nucleotides are not in these MTs. Likewise, it’s not clear to me what the two polymorphic alleles on line 379 are (give locus) and how 83M compares to other strains from Macedonia and Bulgaria. On line 446, it’s stated that only GC2 showed the SNPs, suggesting that this is relative to a reference sequence. What reference is being used here? Is there evidence that GC1 has the ancestral state for these sites? Lines 541-544 state that more strains in databases shared SNPs with the GC1 strains than the GC2 strains, which is confusing because I thought the opposite was stated in the results and on lines 535-536.
- There is a short paragraph, lines 412-415, describing statistics that are not shown in the manuscript. Is there a table missing?
- The in silico analysis of the VNTR loci in other pathovars should be part of the results rather than presented in the discussion. The table summarizing these results should be Table 7 (currently labeled Table 2) and must be cited in the text.
- Line 525: Should be “SNPs shared”
Author Response
We would like to thank the reviewer for the valuable feedback, which has helped to improve the quality of our manuscript. We have tried as best as we could to address all the concerns and we hope that our manuscript is now acceptable for publication in the MDPI journal Microorganisms. Please find our point-by-point response in the attached PDF file.

Reviewer 2 Report
Thorough analysis.
Suggestion: Shorten the Introduction by 25 to 35%. Is there a need to rehash old marker types?
Line 187 (see line 329-330). Figure 1 seems to be missing (or misplaced). A Table with geographic data on collection sites would be useful.
Would Table 1 be better placed in Supplementary info?
Line 325. Should this be Figure 2? Authors should define all the abbreviations used in the figure, in the figure legend. Do figures need to be renumbered?
Why is there a second Table 2 at line 513??
Line 578 and 579. Are these Supplemetary Info (Figure S1 and Table S1) referered to in the text?
Are all those references really necessary?
No discussion of resequencing (low coveage genome sequencing) for strain identification?
Author Response

(The authors gave the same response as above.)
